# Multi-Objective Alignment of LLMs with ORPO using Self-Judgement

## Abstract

The alignment of Large Language Models (LLMs) is achieved through fine-tuning with human preference data, where preference optimization has become a critical part of the process. Many methods have scaled LLM performance by incorporating self-judgement, highlighting the importance of unifying LLM-as-a-judge with the alignment process. One such method, called Self-rewarding LLMs, iteratively samples new data from the model to improve alignment using self-judgement. Since this additional data is generated by the LLM, we argue that similar improvements can be achieved without new data. We propose a method that reuses alignment data in the form of a self-judgement classification task and defines a multi-objective optimization problem. Our self-judgement task is derived from a simple transformation of the primary alignment data, asking the LLM to select the superior response. It does not introduce new data beyond existing alignment data. Thus, we claim that the improvements are due to positive interference between the two tasks. We focus on a direct preference optimization method called Odds-Ratio Preference Optimization (ORPO). We conduct a thorough study of linear scalarization on two objectives and introduce two alternative approaches that vary the emphasis on alignment versus self-judgement objectives. Our results in Mistral 7B indicate a promising direction for fine-tuning LLMs on multiple objectives, particularly to improve performance on related tasks without additional natural language data.

## 1 Introduction

Large Language Models (LLMs) acquire the majority of their linguistic knowledge and capabilities during the pre-training phase, where they learn to predict subsequent tokens from vast amounts of text data, encompassing trillions of tokens Cao et al. (2024). To further refine their abilities and adapt to new tasks, LLMs undergo supervised fine-tuning, a transfer learning method that enhances their performance on downstream applications. One such application is alignment of LLMs.

Alignment of LLMs to human values, and instruction-following is achieved by a combination of supervised fine-tuning and preference optimization. Once an LLM is fine-tuned using supervised labels, additional training on pairs of responses—where one response is preferred over the other—significantly improves alignment performance. This process, known as preference optimization, was first introduced through Reinforcement Learning with Human Feedback (RLHF) on GPT-3 Ouyang et al. (2022). In RLHF, the model is rewarded for generating the preferred response and penalized for the less desirable one, enabling the LLM to align with human values such as helpfulness and truthfulness based on these preferences. A prominent method for achieving preference optimization is Direct Preference Optimization (DPO) Rafailov et al. (2023), which eliminates the need for explicit reward models or reinforcement learning. Instead, DPO directly approximates a loss function based on implicit rewards. One work we use as a baseline further simplifies DPO. Odds Ratio Preference Optimization (ORPO) Hong et al. (2024) integrates preference optimization and supervised fine-tuning into a single loss function using the concept of odds ratio, improving alignment performance while reducing computational requirements by half.

Self-rewarding Yuan et al. (2024) offers a novel approach to enhancing the performance of LLMs on alignment tasks through iterative fine-tuning. In this method, responses are sampled from the LLM after each iteration to generate fresh data. A key innovation in this approach is training the

Figure 1: Alignment data (left) contains instruction and response pairs, with one response being higher quality than the other. Self-judgment data (right) follows the same format and poses a binary classification task using a prompt asking the LLM to pick the higher quality response. This figure shows the relationship between the two datasets.

LLM's ability to self-judge its responses, categorizing them into accepted or rejected groups to create new data. This technique has demonstrated significant improvements, particularly up to the third iteration.

We propose an alternative approach to achieve the data-dependent gains observed in self-rewarding methods Yuan et al. (2024). While self-sampling primarily relies on the LLM's internal knowledge—introducing little new information into the training process—we leverage both self-judgment and alignment data through a multi-objective preference optimization framework. Specifically, we construct self-judgment data by reusing the alignment data with a new prompt, as illustrated in Figure 1. Recognizing the challenges of scaling LLMs with limited external and internal data, we propose an alternative to the self-rewarding process. Instead of performing seed training of an LLM-as-a-judge and generating new data for epoch-wise performance scaling, we employ multi-task learning Hu et al. (2023) to achieve similar improvements. We hypothesize that fine-tuning LLMs on multiple objectives, including self-judgment, can lead to positive interference, achieving some of the benefits of self-rewarding without the need to generate new data. Our approach is grounded in the machine learning paradigms of multi-objective optimization Gunantara (2018) and multi-task learning Hu et al. (2023).

To summarize, we propose a significant direction for improving LLMs by fine-tuning them on multiple related objectives simultaneously, demonstrating better generalization on the main task. With minimal additional data beyond the provided preferences, our approach offers an alternative to iterative fine-tuning improvements.

Our main contributions are as follows:

- We conduct an extensive multi-objective optimization study on the Mistral 7B model, utilizing self-judgment data.
- We provide evidence of positive interference when simultaneously training for alignment and self-judgment objectives.
- We propose a data-constrained method for scaling alignment performance of an LLM through multi-objective optimization.

## 2 RELATED WORK

During pre-training, LLMs acquire the majority of their natural language knowledge and skills by predicting the next token across trillions of tokens Cao et al. (2024). LLMs are adapted to new skills and behaviors through a transfer learning technique known as supervised fine-tuning. This fine-tuning stage differs significantly from pre-training, as it involves making subtle adjustments to the

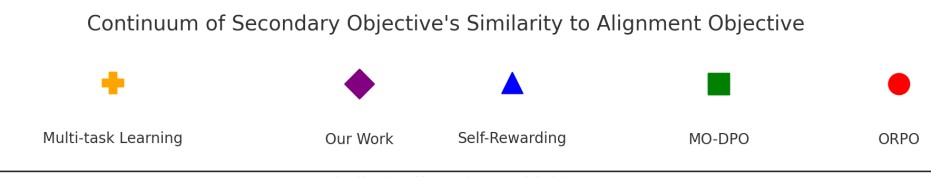

Figure 2: Our self-judgment task, which involves following instructions with a rigid binary output, is somewhat dissimilar to the alignment task. This distinction leads us to categorize it as being closer to Multi-task Learning (MTL) than Self-rewarding LLMs(refff)and Multi-objective DPO Zhou et al. (2023).

LLM's weights using a much lower learning rate. Because of these small changes, the quality of the fine-tuning dataset becomes crucial; relying on labels from weaker models can degrade the LLM's performance Burns et al. (2023). Our work fits into several key areas: preference optimization, multi-objective optimization, and LLM-as-a-judge frameworks. Additionally, it offers an alternative approach to iterative fine-tuning, suggesting new ways to enhance LLM performance without relying on the iterative self-sampling methods.

## 2.1 PREFERENCE OPTIMIZATION

To align pretrained Large Language Models (LLMs) with human intents and teach them to follow instructions, they are fine-tuned on instruction-following datasets. The first example of instruction-following datasets came with the Flan family Chung et al. (2022), which introduced a family of language models that were fine-tuned using supervised fine-tuning (SFT). Supervised Fine-Tuning (SFT) optimizes the log likelihood of the target outputs in the language model. Since then, GPT-3 Ouyang et al. (2022) introduced Reinforcement Learning with Human Feedback (RLHF) on a then-novel kind of data for language models: "human preferences." By providing contrasting examples, GPT-3, along with its vast scale, earned LLMs their reputation as instruction followers. Since then, preference optimization has become a critical component of aligning language models to human intent. Direct Preference Optimization (DPO) Rafailov et al. (2023) provided a significant breakthrough in this process by simplifying preference optimization. They removed Reinforcement Learning components from the pipeline, such as training reward models, and approximated rewards directly in terms of the LLM's probabilities of producing output labels. DPO uses a supervised loss for preference optimization and performs as well as RLHF in most cases. Building on the same principles, Odds Ratio Preference Optimization (ORPO) Hong et al. (2024) further simplifies the pipeline by removing the reference LLM, and by combining SFT and preference optimization into a single loss function made of multiple objectives. Due to its efficiency benefits over DPO, we use ORPO as our baseline.

## 2.2 ITERATIVE FINE-TUNING

Iterative fine-tuning Yuan et al. (2024); Chen et al. (2024) achieves stellar scaling of performance across various tasks, including alignment. By generating new data, either as preferences or as both instructions and preferences, iterative fine-tuning efforts strive to improve LLMs continuously. However, they rarely show any improvements after three iterations, and the creation of new data is an expensive ordeal. One notable work that we provide an alternative to is self-rewarding Yuan et al. (2024). They train an LLM across two dimensions: LLM-as-a-judge and alignment to human preferences. This is very similar to what we do with our multi-objective ORPO Loss. This contrasts with other works that utilize the RLHF pipeline and use LLM-as-a-judge as part of the reference model Lee et al. (2024). In both works, LLM-as-a-judge and alignment objectives are initially fine-tuned without preference optimization, and no improvements in alignment performance are reported during this stage. Only after performing self-sampling iterations with preference optimization—where both chosen and rejected responses are utilized—do these approaches show improvements in alignment performance. In contrast, during supervised fine-tuning, only the chosen responses are used to adjust the model. Our approach eliminates the need for their self-sampling stages by simulta-

neously applying preference optimization and supervised fine-tuning to both the self-judgment and alignment tasks using ORPO.

## 2.3 LLM-AS-A-JUDGE

It's extremely expensive to collect human preference data. Replacing humans with LLMs to act as judges would automate the alignment process. However, fine-tuning LLMs to be judges does not come without its challenges. Various biases, such as verbosity bias, position bias, and format bias, hinder the wide adoption of LLM-as-a-judge Thakur et al. (2024); Zheng et al. (2023). Position bias causes the judges to prefer the first-mentioned response with more probability Shi et al. (2024), while verbosity bias occurs when LLMs tend to prefer verbose responses over succinct and accurate ones. LLM-as-a-judge models are trained to produce elaborate explanations and scores to judge responses. Our work only uses pairwise selection tasks without using the explanations, which is similar to On-policy self-judgment Lee et al. (2024). Self-rewarding is also inspired by LLM-as-a-judge and enhances their prompt to elicit scores with a breakdown of important components. This simple innovation proves pivotal as their approach is unable to harness similar improvements without the prompt enhancements.

## 2.4 MULTI-OBJECTIVE AND MULTI-TASK LEARNING IN LLMS

Recent advancements have seen a surge in the use of multi-objective preference optimization for large language models (LLMs). Methods like Multi-Objective DPO (MODPO) Zhou et al. (2023) adapt the original DPO approach into a multi-objective problem using techniques akin to linear scalarization Hu et al. (2023). Other studies integrate multiple objectives within the alignment task Wang et al. (2024); Guo et al. (2024), focusing on factors like helpfulness, truthfulness, honesty, and instruction adherence. These methods manage trade-offs between objectives, ensuring the LLM operates on the Pareto front where no other optimization outperforms it.

However, as illustrated in Figure 2, our work differs by introducing a separate task—self-judgment—that is significantly distinct from the original instruction-following task. While existing multi-objective (MO) methods aim to align the model across different sub-goals within the same task, our inclusion of self-judgment data broadens the scope, placing our work firmly within the multi-task learning (MTL) framework. In MTL, the objective is to learn shared representations across multiple tasks to enhance generalization through positive interference. Although MTL often grapples with challenges like negative interference due to gradient conflicts Shi et al. (2023); Yu et al. (2020); Sener & Koltun (2018), we employ MO optimization techniques as a method to achieve effective MTL without the computational overhead of mitigating these conflicts.

## 3 METHOD

### 3.1 PRELIMINARIES

**Pareto Optimality in Multi-objective Optimization**

Pareto optimality Miettinen (1999) is a key concept in multi-objective optimization, where the goal is to optimize several objectives simultaneously. A solution is said to be Pareto optimal if no objective can be improved without causing a deterioration in at least one other objective. In mathematical terms, consider a multi-objective optimization problem with $k$ objective functions:

$$\min_{x \in X} F(x) = (f_1(x), f_2(x), \ldots, f_k(x))$$

where $F(x)$ is a vector of objective functions $f_1, f_2, \ldots, f_k$, and $X$ is the feasible set of solutions. A solution $x^* \in X$ is said to be Pareto optimal if there is no other solution $x \in X$ such that:

$$f_i(x) \leq f_i(x^*) \quad \forall i \in \{1, 2, \ldots, k\}$$

with at least one strict inequality:

$$f_j(x) < f_j(x^*) \quad \text{for some } j \in \{1, 2, \ldots, k\}$$

This condition means that it is impossible to improve the value of any one objective function without worsening the value of at least one other objective. The set of all Pareto optimal solutions forms what is known as the Pareto front, a surface that represents the trade-offs between conflicting objectives. Formally, the Pareto front $P$ can be defined as:

$$P = \{x^* \in X \mid \nexists x \in X \text{ such that } f_i(x) \leq f_i(x^*) \, \forall i \text{ and } f_j(x) < f_j(x^*) \text{ for at least one } j\}$$

We search for such non-dominated solutions using linear scalarization and conditional optimization techniques.

**Odds-Ratio Preference Optimization**

We use Odds-Ratio Preference Optimization (ORPO) Hong et al. (2024) as our baseline. ORPO simplifies the preference optimization process by replacing the reference LLM with an odds ratio. The odds are calculated by incorporating the log probabilities of an output $y$ given an input prompt $x$, as shown in equation (1). This forms a self-ratio that increases the likelihood of the chosen response over the rejected response. The odds ratio between the winning (chosen) response $y_w$ and the losing (rejected) response $y_l$ is defined in equation (3).

$$\log P_\theta(y|x) = \frac{1}{m} \sum_{t=1}^{m} \log P_\theta(y_t|x, y_{<t}) \tag{1}$$

$$\text{odds}_\theta(y|x) = \frac{P_\theta(y|x)}{1 - P_\theta(y|x)} \tag{2}$$

$$L_{\text{OR}} = -\log\left(\sigma\left(\log \frac{\text{odds}_\theta(y_w|x)}{\text{odds}_\theta(y_l|x)}\right)\right) \tag{3}$$

Supervised Fine-tuning (SFT) is incorporated into the process, as both preference optimization and SFT are performed simultaneously, as shown in equation (4). The ORPO paper employs a $\lambda$ value of 0.1, which we also adopt in our approach.

$$L_{\text{ORPO}} = \mathbb{E}(x, y_w, y_l)\left[L_{\text{SFT}} + \lambda \cdot L_{\text{OR}}\right] \tag{4}$$

## 3.2 MULTI-OBJECTIVE OPTIMIZATION

Our approaches fall into three categories. The first is linear scalarization, which computes a weighted sum of two loss functions. We conduct a series of experiments with different scalarization weights to evaluate its impact. The second approach, conjoined, trains alignment odds in relation to self-judgment odds, leading to rapid improvements in alignment response odds. In contrast, the third approach, conditional, reduces the emphasis on self-judgment training when its odds rise above zero, preventing overfitting on the simpler self-judgment task.

### 3.2.1 LINEAR SCALARIZATION

We take a weighted sum approach to calculate a multi-objective loss between the alignment and self-judgment tasks.

$$L_{\text{MO}} = w_1 \cdot L_{\text{Alignment}} + (1 - w_1) \cdot L_{\text{Self-Judgment}} \tag{5}$$

Both $L_{\text{Self-Judgment}}$ and $L_{\text{Alignment}}$ are $L_{ORPO}$ losses, as shown in Equation (4). They are calculated for the self-judgment and alignment data from Fig. 1, respectively.

### 3.2.2 CONJOINED MULTI-OBJECTIVE LOSS

Inspired by works like SPIN Chen et al. (2024), which pits LLMs against themselves, we create a conjoined ORPO loss. We scale the implicit rewards of the alignment task using the frozen odds ratio of the self-judgment task $\theta'$ generated at the same step. This propels higher gradients for the alignment task, as it is expected to always be higher than the self-judgment task's odds ratio.

$$L_{\text{OR}} = -\log\left(\sigma\left(\log\frac{\text{odds}_\theta(y_w|x)}{\text{odds}_\theta(y_l|x)} \cdot \frac{\text{odds}_{\theta'}(y_l|x')}{\text{odds}_{\theta'}(y_w|x')}\right)\right) \tag{6}$$

**Gradients of Conjoined methods** If we attribute frozen odds ration from equation (6) as $k$, we see how the gradients are scaled by the first term shown in the below equation.

$$\nabla_\theta \mathcal{L}_{OR} = \left(1 + k \cdot \frac{\text{odds}_\theta P(y_w|x)}{\text{odds}_\theta P(y_l|x)}\right)^{-1} \cdot$$
$$\nabla_\theta \log\left(\frac{\text{odds}_\theta P(y_w|x)}{\text{odds}_\theta P(y_l|x)}\right)$$

The gradients will be higher when $k$ is lower, and vice versa, assuming all other factors remain constant.

### 3.2.3 CONDITIONAL MULTI-OBJECTIVE OPTIMIZATION

In e-constraint method Pirouz & Khorram (2016), one of the objective functions is optimized, while the remaining objectives are treated as constraints, bounded by certain threshold values (denoted by $e_i$). Given $k$ objective functions $f_1(x), f_2(x), \ldots, f_k(x)$, the method seeks to optimize a single objective (say $f_1(x)$) subject to constraints on the other objectives:

$$\min_{x \in X} f_1(x) \quad \text{subject to} \quad f_i(x) \leq e_i \quad \forall i = 2, 3, \ldots, k$$

Our Conditional Multi-objective Optimization Algorithm is a rendering of e-constraint method.

---

**Algorithm 1** Conditional Multi-objective Optimization Algorithm

1: **Input:** $\log\frac{\text{odds}_{\theta,\text{SJ}}(y_w|x)}{\text{odds}_{\theta,\text{SJ}}(y_l|x)}$, num_epoch, $L_{\text{Alignment}}$, $L_{\text{Self-Judgment}}$, $w_1$, $w_2$
2: **Output:** loss
3: **if** $\log\frac{\text{odds}_{\theta,\text{SJ}}(y_w|x)}{\text{odds}_{\theta,\text{SJ}}(y_l|x)} < 0$ **then**
4: $\quad loss \leftarrow (w_1 \times L_{\text{Alignment}} + w_2 \times L_{\text{Self-Judgment}})$
5: **else**
6: $\quad loss \leftarrow L_{\text{Alignment}}$ ▷ Only optimize alignment objective if SJ odds are good
7: **end if**

---

Due to difference in convergence rates, and complexity of self-judgement and alignment data, we investigate a conditional training on Self-judgement loss. When the odds of the winning response are lower than losing response in Self-judgement data, we activate the self-judgement ORPO Loss as shown in Algorithm 1.

## 4 EXPERIMENTS AND RESULTS

**Baseline and Setup:** We conduct our experiments on Mistral 7B Jiang et al. (2023) using the Ultra-binarizedFeedback dataset Tunstall et al. (2023). We filter the dataset by keeping the total length of both prompts and the two responses under roughly 5,500 tokens, as this helps us maintain a 6k total limit on self-judgment prompts. This filter retains roughly 80% of the full dataset. We measure our results on Alpaca Eval, comparing them against the Meta LLama 3 70B model. We employ a learning rate of 5e-6 with cosine scheduling for the duration of 3 epochs across all our experiments. We attempted higher learning rates, such as 9e-6 for multi-objective experiments, but this led to over-fitting—i.e., training loss was minimized, but performance on Alpaca Eval significantly worsened. All the reported results use length-controlled Alpaca Eval win-rates, and thus are representative of the quality of responses independent of the length of the response. We present the best results from either epoch 2 or epoch 3. Each of the tables shows which epoch the reported results are taken from. In all cases, the results from epoch 2 and epoch 3 are quite similar for a given experiment.

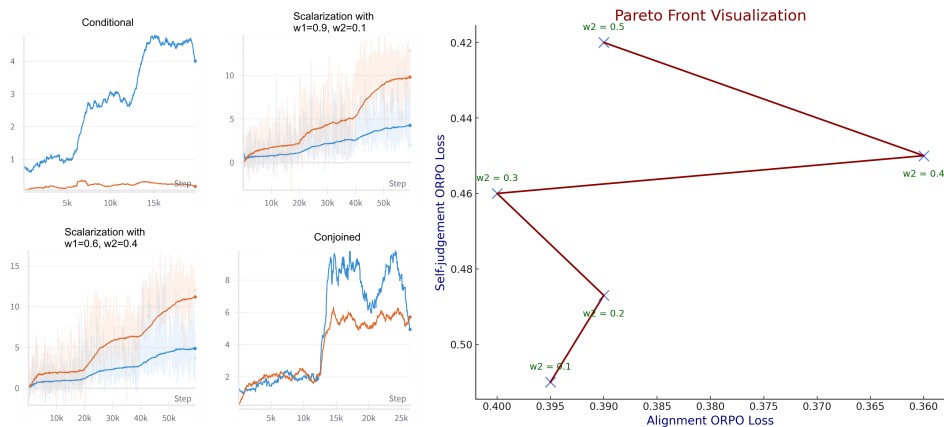

Figure 3: (left) Comparison of Log odds (y-axis) along optimizer steps (x-axis) of Alignment Task (blue) and Self-judgment Task (orange) are shown for various experiments. (right) Pareto front of non-dominated solutions is shown. For the training loss, we see how $w1 = 0.6$ strictly dominates some other points which is reflected in it's final results.

**Self-Judgment Prompt and Biases** We use a basic prompt as shown in Figure 1. However, we observed prompt leakage when using just one prompt. To address this, we sampled five diverse main prompts from ChatGPT and LLM-as-a-judge Shi et al. (2024); Thakur et al. (2024); Zhu et al. (2023) and made minor tweaks to variables such as response names and the order of instructions, creating approximately 50 prompt variations to overcome prompt leakage and format bias. These changes resolved the prompt leakage issue. Additionally, we created an ordering where the first response is the rejected response 55% of the time to address position bias in LLMs, which tend to pick the first response as better more often. This strategy effectively tackled position bias.

### 4.1 Multi-objective weighted Sum Results

Our best results were obtained with $w1 = 0.6$, as shown in Table 1. However, the most informative results came from the $w1 = 0.8$ and $w1 = 0.9$ runs, where we observed the worst performance. We believe this is due to gradient interference between the two objectives. This suggests that, in scenarios with low weight on self-judgment, the task may not be adequately optimized and could act as noise during the initial stages of training. By the end of training, we observe that the odds ratios for alignment and self-judgment converge, as shown on the right side of Figure 3.

From the results in Table 1, we constructed the Pareto front for the training loss, depicted on the right side of Figure 3. We notice that the Pareto front partially predicts the performance on the Alpaca Eval benchmark, with the best results emerging from the $w1 = 0.6$ and $w1 = 0.7$ runs. However, the training loss for the alignment task was also significantly minimized in the $w1 = 0.5$ and $w1 = 0.8$ runs. Our Pareto front figure further shows that the $w1 = 0.6$ run strictly dominates both the $w1 = 0.7$ and $w1 = 0.8$ runs, indicating that these points do not lie on the true Pareto front.

| Model | w1 | w2 | Epochs | WR |
|---|---|---|---|---|
| baseline | 1.0 | 0.0 | 3 | 18.72 |
| Scalarization | 0.9 | 0.1 | 3 | 16.70 |
| Scalarization | 0.8 | 0.2 | 2 | 16.32 |
| Scalarization | 0.7 | 0.3 | 3 | 18.76 |
| Scalarization | 0.6 | 0.4 | 2 | **19.72** |
| Scalarization | 0.5 | 0.5 | 2 | 17.20 |

Table 1: Table for Multi-objective weighted approach on Alpaca 2 results against Llama 3 70B. WR stands for length controlled Winning Rates.

| Model | w1 | w2 | Epochs | WR |
|---|---|---|---|---|
| baseline | 1.0 | 0.0 | 3 | 18.72 |
| conjoined | 0.7 | 0.3 | 2 | 18.03 |
| scalarization | 0.7 | 0.3 | 3 | 18.76 |
| conditional | 0.9 | 0.1 | 3 | **19.55** |
| epoch-by-epoch | 1.0 | 1.0 | 3.5 | 14.84 |
| task-by-task | 1.0 | 1.0 | 2 | 12.5 |
| scalarization (SJ-SFT) | 0.6 | 0.4 | 3 | 13.1 |

Table 2: Table for comparison of multi-objective approaches on Alpaca Eval against Llama 3 70B

## 4.2 APPROACH BASED COMPARISON

Since we observed a rapid increase in self-judgement (SJ) log odds during scalarization experiments (as shown in Figure 3 right), we concluded that the model overfits the SJ data due to its simplicity or underfits the alignment task by comparison. To improve the fit on the alignment task, we employed the conjoined approach; to reduce overfitting on the self-judgement task, we used the conditional approach.

These two methods are based on conflicting hypotheses. Our results indicate that the conditional hypothesis is correct. Specifically, our linear scalarization models overfit the SJ data while fitting the alignment data reasonably well. In the conjoined approach, the loss for alignment data will be higher if their probabilities are lower than those for the easily maximized SJ data. Therefore, the conjoined method leads to very high probabilities for chosen responses compared to baseline and linear scalarization approaches. For example, we found that the odds ratio for conjoined alignment examples exceeds 50, whereas the same ratio remains below 7 for scalarization experiments, as shown in Figure 3 (left).

Our approach-wise results in Table 2 demonstrate that the epoch-by-epoch and task-by-task strategies, where we alternated between alignment and self-judgement ORPO training, performed significantly worse than the baseline. We believe this is due to the use of longer context lengths and the clash between the simplicity of the binary self-judgement task and the complexity of the alignment objective. However, the low win rate in task-by-task training suggests that explanations used in the self-rewarding LLM-as-a-judge framework are crucial for maintaining alignment performance. Nonetheless, our simultaneous training approach exhibited less performance degradation, likely due to positive interference between the two tasks and the control offered by assigning extra weight to the alignment task.

We consider scalarization (SJ-SFT), applied epoch-by-epoch, as control experiments that justify the superior results of the scalarization and conditional approaches. By replacing the ORPO loss for SJ data with supervised fine-tuning on chosen data (SFT), we observe considerable performance degradation. This indicates that we have incurred overfitting due to the usage of the same data for both SJ and alignment tasks. Through simultaneous training, we avoid such performance degradation caused by data duplication. This further indicates the importance of our results in seeking simultaneous scaling of two differing objectives in the multi-task learning framework. We observe that careful tweaking of their importance results in better-than-baseline performance.

## 4.3 VERBOSITY BIAS

All of our multi-objective experiments (with the exception of conjoined method) tend to generate longer responses compared to the baseline, which diminishes the potential benefits of our approach. We attempted to tackle this with prompting, but it caused no change in the final win-rate. We attribute this to two factors:

- Padding used for self-judgment data: Since self-judgment data contains very lengthy prompts, and LLMs are usually trained with a consistent length for the entire dataset for efficiency, padding is necessary. For self-judgment data, padding is more abundant than for the baseline. In our results, we observe that this impacts the average length of the trained models' outputs on Alpaca Eval. There is no direct way to address this issue, as LLMs require padding during training.

- Verbosity Bias from LLM-as-a-judge training: Verbosity serves as a useful heuristic for predicting winning responses over losing ones. LLM-as-a-judge learns quickly to rely on such measures to reduce training loss. This trap is the biggest challenge we faced, and we have not found a consistent solution for it.

## 5 CONCLUSION AND FUTURE WORK

Our results indicate a promising direction for fine-tuning LLMs. The reliance on high-quality data to scale LLM performance and achieve AGI will eventually reach a ceiling limited by human performance. Therefore, methods like multi-objective optimization inspired by human learning, are needed to scale LLMs without relying on additional high-quality data. Our self-judgment task provided no extra information beyond what was already available in the dataset; rather, it aimed to constrain the solution space to a subset where two objectives are satisfied instead of one. This also introduced unhelpful noise in form of verbosity. We believe that future work will unlock the full potential of this approach by addressing the tendency of multiple objectives to interfere and fit to noise. Perhaps, we may see improvements in performance that surpass what self-rewarding LLMs have achieved with their iterative fine-tuning approach.

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
