# OpenReview forum: "Multi-Objective Alignment of LLMs with ORPO using Self-Judgement"
_ICLR.cc/2025/Conference — Submitted to ICLR 2025_

### Official Review · Reviewer_sjFv · 2024-11-03

**Soundness:** 2
**Presentation:** 1
**Contribution:** 1
**Rating:** 3
**Confidence:** 4

**Summary:**

This paper proposes a multi-objective alignment method based on ORPO. More specifically, the first objective is the same as the log odds alignment objective proposed in the ORPO paper, the authors propose to add a second objective where the LLM should predict which response is preferred by humans. They test their approach on Alpaca Eval and ablate different design decisions.

**Strengths:**

The ablation study of the proposed method appears to be thorough. The authors systematically compare various strategies for combining the two objectives, including conditional, epoch-by-epoch, and task-by-task approaches, and report detailed results for each.

**Weaknesses:**

This paper would benefit from substantial improvements across several dimensions:

**Presentation:** The framing of the paper is somewhat misleading. The primary contribution appears to be the proposal of combining two objectives and exploring various ways to integrate them. However, certain claims, such as the statement on line 97, position this work as an alternative to iterative fine-tuning. The intent of sampling from the current policy in iterative fine-tuning is generally to produce more on-policy data, which can help bridge the performance gap between offline and online methods. I would accept this framing if the paper presented a novel approach for obtaining on-policy data without direct sampling from the policy itself. As it stands, this framing seems to overstate the contribution.

**Motivation:** The motivation lacks clarity. The authors suggest that because LLMs are often used to judge their own outputs, sampling from the model may not be necessary. This argument is unclear; it's possible that verification tasks are fundamentally easier than generation tasks—an LLM may perform well as a judge but poorly as a generator, thereby effectively evaluating its own inputs and improving performance. Additionally, the paper does not seem to empirically test this hypothesis. If the aim is to demonstrate that the proposed approach can replace iterative fine-tuning, then a direct comparison with iterative fine-tuning methods is essential, showing that similar or better performance can be achieved with comparable or less data.

**Experiments:** The experimental design has some gaps. Key baselines, such as the performance of the supervised fine-tuning (SFT) model or the pretrained model, are missing. To evaluate the effectiveness of the proposed method, comparisons with established alignment methods like Direct Preference Optimization (DPO), Proximal Policy Optimization (PPO), and other conventional techniques are necessary.

**Questions:**

- What is the y-axis in Figure 3?
- Line 283 refers to equation 8, but I don't see any equation 8
- In equation 6, $\theta'$ is said to be frozen weights of the self judgement task, it is the parameters of the same model, right? Also, in the same equation, shouldn't you use a different prompt for this task and the alignment task? (it seems that both tasks are using $x$)
- Why is the second task called self judgment in the context of your work? It doesn't seem to be that you are generating the responses from the same LLM, so it is not really **self** judgement, is it?

---

> ### Author Response · Authors · 2024-11-27
>
> Thank you for your valuable feedback. We are grateful of your time invested in improving the quality of this research.
>
> We have taken each of your feedback into account, and edited the paper in the following manner:
>
> 1. Our method is not a direct alternative to iterative finetuning. It was inspired by iterative finetuning but it does not directly compare with it. We use iterative finetuning as an argument that simultaneous training could be beneficial, and we prove that with our results that it is in some cases. Therefore, we have corrected our contribution claim 3 to “We propose a data-constrained method for scaling alignment performance of an LLM through multi-objective optimization.”
>
> 2. Regarding motivation; In self-rewarding work we reference, they use the LLMs to generate new on-policy data in form of instructions and responses and they use LLM’s improved self-judgement capability from a seed training set to provide preference labels for the same data. We argue that if LLM is simultaneously trained for self-judgement and alignment, this slow interaction based on generating new data across epochs, would be replaced with Multi-task learning setup by using Multi-objective optimization. It is not a direct replacement of iterative finetuning as you said. But we try to draw from it to justify why positive interreference was expected between the two tasks.  The clarification is added to line 077 para.
>
> In experiments, we agree that we did not run Multi-objective experiments where Self-judgement data is merely training by SFT. However, we have ran those experiments after taking feedback from the reviewers. Our Multi-objective results with ORPO loss on alignment, and SFT loss on Self-judgement(SJ) data gives 13.1% win rate as compared to MO with ORPO loss for both alignment and SJ data. We found it slightly surprising because, one of our hypothesis was that ORPO might not be a good fit for SJ data, but our results claim otherwise.
>
>
> To answer each of your questions:
>
> Questions:
> • "What is the y-axis in Figure 3?" It refers to optimizer Steps. We have added the information to the figure description as well.
> • "Line 283 refers to equation 8, but I don't see any equation 8." Apologies, we had edited that equation out and forgot to edit out the line.
> • "In equation 6, θ′ is said to be frozen weights of the self judgement task, it is the parameters of the same model, right? Also, in the same equation, shouldn't you use a different prompt for this task and the alignment task? (it seems that both tasks are using x) " Thanks for the corrections. We have made the adjustments to both input and prompt for self-judgement as x', and y'.
> • "Why is the second task called self judgment in the context of your work? It doesn't seem to be that you are generating the responses from the same LLM, so it is not really self judgement, is it?" It is because we are training it to self-judge. Some other works have also taken a very similar SJ prompt as us, and used the same key word. For example, “Aligning large language models by on-policy self-judgment” paper.

---

### Official Review · Reviewer_LM5Q · 2024-11-03

**Soundness:** 2
**Presentation:** 2
**Contribution:** 2
**Rating:** 3
**Confidence:** 4

**Summary:**

The authors propose a new alignment method for LLMs that uses multi-objective optimization to combine (1) self-judgment and (2) odds-ratio preference optimization (ORPO; Hong et al. 2024).  This paper explores a few general techniques for performing multi-objective optimization for these two objectives.  They present some promising preliminary experiments.

**Strengths:**

Exploring techniques for combining the strengths of various learning paradigms through multi-objective optimization is an interesting avenue to explore.  Similarly, looking at methods to exploit LLMs as judges is also interesting.

**Weaknesses:**

I found this paper hard to follow.  The clarity of the presentation and overall pitch of the paper needs to be improved.

The method's empirical validation is weak. I do not see a meaningful comparison to other methods.

I do not see a good case for why the paper needs to combine specifically ORPO with LLMs-as-judge.  What about other competing alignment objectives?

**Questions:**

What happens if you replace ORPD with a more popular objective like DPO?  Are you open to adding this comparison to the paper?

Multi-objective optimization techniques should support more than two objectives.  Why did you limit the experiments to just two objectives?

---
Suggestions:

 - Please learn the difference between textual (`\citet`) and parenthetical (`\citep`) citations.

 - I suggest moving the long related work section to a later section of the paper or an appendix.  This will help refocus the paper in my opinion.

---

> ### Author Response · Authors · 2024-11-27
> **Response to the questions**
>
> Thank you for your valuable feedback. We are grateful of your time invested in improving the quality of this research.
>
> Response to the weaknesses:
>
> "I do not see a good case for why the paper needs to combine specifically ORPO with LLMs-as-judge. What about other competing alignment objectives? "
> We combine LLM-as-a-judge training with alignment training. We choose ORPO because of its efficiency in training for preference optimization. There are many other choices like DPO, SimPO etc. By creating a simultaneous training of self-judgement and alignment objective, we recreate the benefits that the self-sampling steps of iterative finetuning produce in works like self-rewarding LLMs. This is the main idea of the paper.
>
> Response to the questions:
>
> "What happens if you replace ORPO with a more popular objective like DPO? Are you open to adding this comparison to the paper?"
> We considered it at first. But DPO takes 4 times more time to run than ORPO. And ORPO improves DPO in performance as well as efficiency. So, we decided to focus our work on ORPO. We believe, whatever works for ORPO, should work for DPO as well.
>
> "Multi-objective optimization techniques should support more than two objectives. Why did you limit the experiments to just two objectives?"
> First reason is that it would be highly computationally demanding. Secondly, introduction of the second objective is based on a series of works on iterative finetuning and LLM-as-a-judge fields. We are not aware of any other task that can be constructed from the alignment data (apart from self-judgement). Self-judgement has a unique structure with respect to preferences data, as you can duplicate it without any extra information. Please note that, we do not refer to branches of alignment data like helpfulness, truthfulness etc. as separate multi-objective dimensions for our current work since we treat them as one task. Those studies are different from the scope and main idea we presented. You are correct that Multi-task learning and MOO usually train multiple, not just 2 tasks at once. However, it is beyond the scope of our work to discuss and find more such dimensions.

---

### Official Review · Reviewer_azDC · 2024-11-04

**Soundness:** 1
**Presentation:** 3
**Contribution:** 2
**Rating:** 3
**Confidence:** 4

**Summary:**

The paper proposes multi-objective optimization of self-judged preference data and available alignment data. The self-judged data is collected on a dataset where the human annotated preference labels are already available. The authors propose that in addition to optimizing the model on annotated preference data we should also optimize on self-judged preference data. The authors use two multiobjective optimization algorithm on these two objectives. Additionally they use one algorithm called CONJOINED MULTI-OBJECTIVE LOSS where they optimize against the self-judgement objective. The experiment result does not show any significant gains compared to the baselines.

**Strengths:**

1. The authors clearly explain their approach and it's connection to previous approaches. The novel contributions have been clearly explained.
2. For self-judgment data collection the authors have addressed the issue of various types of biases. To mitigate the biases they have created about 50 prompts to get the correct judgement as per the model.

**Weaknesses:**

1. On Figure 3 (right) the Pareto frontier is visualized for different scalar weights. The solutions be smoothly moving on the multiobjective-loss space as the weights are changed. However, the solutions are all over the place. This indicates that the optimization have not been performed well or the random effects (of batch order etc) are too high. This puts into question the validity of the results from the first set of results. I would suggest authors perform thorough optimization through longer training. Otherwise, the authors should perform multiple runs with the same weight and show the averaged results.
2. The proposed approach relies on annotated alignment data. This data already has the gold annotations. Using a small LLM (Mistral-7B) to annotate this again (ie self-judged data) and using this annotation as a separate ground truth does not make sense. The model may just be wrong on it's self-judgement. Unless this self-judgement annotations are collected on a large scale in an unsupervised fashion (as done on the paper Self-Rewarding Language Models), we cannot rely on this self-judgement annotations.
3. The authors have not clearly explained why the two objectives (performance on alignment data and performance on self-judged data)  may be contrary to each other. One way to show that would be to provide the loss for self-judged data for the baseline model. If better performance on the baseline means better performance on self-judged data than the two objectives are always aligned and there is no point in doing multi-objective optimzation.
4. For the CONJOINED MULTI-OBJECTIVE LOSS, the self-judgement objective is optimized against. This is a direct contradiction to the multi-objective optimization that is being argued by the authors.

**Questions:**

1. Can you explain why log-sigmoid is used in equation 3?
2. Is there any numerical stability issues during training? As odds ratio are calculated it to require the probabilities which can be very small and cause numerical stability issues. Does it happen?

---

> ### Author Response · Authors · 2024-11-27
> **response to weaknesses and questions**
>
> Thank you for your valuable feedback. We are grateful of your time invested in improving the quality of this research.
>
> Thank you for your astute observations, however we disagree with some of the weaknesses’ assessments. Here are our responses:
> Weakness 1: We have conducted longer training, as our results are reported after 2-3 epochs. Performance starts to degrade after 3 epochs, in some cases even after 2 epochs. That is why all reported results are from 2-3 epochs of training.
> Weakness 2: We do not annotate separate ground truth for SJ data. Since we know which response is better based on alignment ground truth. And if we create two options based on the choices, we know which one is the superior response for the SJ data as well. Therefore, we do not sample new self-judgement data. We construct it, with a simple prompt and a simple algorithm on the alignment data. For example, alignment data rewards the better response A and penalizes response B. Self-judgement rendering of the same, rewards I choose A, and penalizes I choose B responses.
> Weakness 3: We do not believe the two objectives are contrary to each other. But, to answer your question, when we performed conditional experiments, that improve Multi-objective results, we find that unless SJ loss is trained, it is not affected by the alignment loss. Therefore, we know that SJ loss naturally does not decrease or increase just by training on alignment loss.
> Weakness 4: Conjoined loss is based on the hypothesis that probabilities of alignment responses should be higher than probability of SJ responses. Therefore, roughly speaking we increase alignment probabilities with respect to frozen probabilities of SJ responses. This is not contradictory to the main objective. It tries to radically improve alignment probability and thus tries something unique and is different from other methods we tried.
>
> To answer your questions:
>
> 1. "Can you explain why log-sigmoid is used in equation 3?"
> log-sigmoid setup is pretty common in preference optimization methods such as DPO and ORPO. DPO paper derives it from bradley terry model. Intuitively, it creates a more continued penalty than just logarithmic function, causing improvements even in cases when starting probabilities may be imbalanced.
> 2. "Is there any numerical stability issues during training? As odds ratio are calculated it to require the probabilities which can be very small and cause numerical stability issues. Does it happen?"
> Indeed, ORPO loss is prone to instability. We had to use some code improvements to make it stable, as even small noise in loss definition in pytorch would cause NAN errors. But once our setup started operating smoothly, we did not encounter any instabilities in any of the experiments.

---

> > ### Comment · Reviewer_azDC · 2024-11-30
> > **Response to rebuttal**
> >
> > I thank the authors for their response. I have increased the score to 3 as I believe this paper still needs more work before it can be accepted.

---

### Official Review · Reviewer_tkTp · 2024-11-04

**Soundness:** 3
**Presentation:** 3
**Contribution:** 2
**Rating:** 5
**Confidence:** 3

**Summary:**

The paper proposes a method to fine-tune LLMs by reusing existing alignment data for self-judgment tasks, eliminating the need for self-sampling. This approach utilizes the ORPO method as the primary baseline for supervised fine-tuning. The training process is approached as a multi-objective optimization problem, with comparisons among scalarization, conjoint, and conditional optimization methods. Experiments are conducted on the Mistral 7B model, benchmarked against the Llama 3 70B model on the Alpaca Eval dataset. The results show mixed to slight improvements in alignment in contrast to the ORPO baseline fine-tuned solely on the alignment task.

**Strengths:**

- The integration of ORPO with a multi-objective optimization framework to simultaneously address alignment and self-judgment tasks is innovative and promises efficiency improvements in LLM fine-tuning.
- The introduction, related work and methodology sections are clearly written, providing a solid understanding of the theoretical background and operational framework. This helps in appreciating how the proposed method diverges from and potentially improves upon existing techniques.

**Weaknesses:**

- The experiment and results section reads a bit rushed and lacks examples that reflect the discussed issues such as verbosity and overfitting.  These could be included in an appendix for reference.
- The findings are limited by the use of only one model and one evaluation dataset. Expanding the testing to include models of varying sizes could better determine the method's general applicability and robustness.
- The identification of verbosity as a significant issue without an applicable solution reduces the potential utility of the proposed approach. More testing on prompt engineering or response format control could help mitigate the verbosity bias ( see questions for more details).

**Questions:**

1. The paper tests the proposed method only on the Mistral 7B model and on one evaluation dataset. I understand that finetuning and evals can be costly, testing on other models within the 2B-13B size range could help determine if verbosity is inherent to the method itself or specific to Mistral 7B model?
2. With verbosity identified as a major issue, have you tried prompt engineering that could effectively reduce the bias in model responses, such as asking the model to respond briefly or follow a specific response format?
3.  Regarding the overfitting issue, would separating the alignment dataset into an alignment subset and another non-overlapping self-judgment subset mitigate this problem?
4. Have there been tests where only self-judgment training was conducted (w1=0, w2=1). As a contrast to w1=1, w2=0? I understand that this would not fine-tune on alignment data, but only on model’s reasoning why it chose A over B (or vice versa).
5. It is not clear how a longer context length might be an issue. The example in Figure 1 shows minimal differences between the two cases, as the self-judgment dataset wraps a rather brief prompt around the rather lengthier alignment data.
6. It is not clear why some training runs have 2 epochs while others have 3.
7. By equation 8, which is missing, do you mean equation 7 for the gradient of Conjoined method?

---

> ### Author Response · Authors · 2024-11-27
> **Response to questions**
>
> Thank you for your valuable feedback. We are grateful of your time invested in improving the quality of this research.
>
> To answer your questions:
> 1. The verbosity could be specific to Mistral 7B. However, not all Mistral 7B models like the baseline fall into the trap of verbosity. This indicates it might not be specific to Mistal 7B. Also, we considered running the same experiment on 2B models, but their context length is less than 4k, which limits the usage of data.
> 2. We attempted prompting to deal with verbosity bias based on your suggestion but it gave similar results. For example, the best MO attains 19.6% win rate without any verbosity prompt. But with a prompt, we get 19.3% . Similarly, it also gets very similar results for baseline. We used the prompt: “Provide a short and focused response in 200 words or less.”
> 3. Self-rewarding takes this approach you described. However, on-policy self-judgement paper that we draw upon does not. We chose to use the same data twice because we want to achieve positive interference for each example between the two tasks. And using complimentary data would result in gradients from two different examples for the two different tasks, increasing the chance of negative interference. We observe the overfitting you described in task-by-task and epoch-by-epoch experiments.
> 4. We have not conducted w1=0, w2=1 experiments. But we did conduct various permutations of w1=1, w2=0 for 1 epoch and then w1=0 and w2=0 for next epochs. However, each resulted in considerably lower win rates as shown in Table 2.
> 5. Context length is considerably longer due to the two added responses, in addition to the prompt. You are right that the prompt itself is quite small, but the two responses add up to as long as 6000-8000 tokens. However, we cap the maximum context length to 6k, and filter our data such that SJ prompt+ 2 responses do not exceed 6k length.
> 6. We take the best of the two epochs. Usually, epoch 2 and 3 have identical results. In rare cases, epoch 3’s performance is slightly worse than epoch 2’s saved model. We have added this clarification in the baseline and setup para for the results section.
> 7. Thanks for the correction.

---

### Official Review · Reviewer_2Rx7 · 2024-11-04

**Soundness:** 2
**Presentation:** 2
**Contribution:** 1
**Rating:** 3
**Confidence:** 3

**Summary:**

This work presents an approach of multi-objective alignment of LLMs for both answering and self-judging abilities. Specifically, the model is tuned with both aligning and self-judgment objectives, and various ways to combine the two objectives are investigated. The tuning itself does not require sampling new data, which makes the process data and computational efficiency. Experiments on Alpaca Eval show the effectivenss of the proposed method in some way.

**Strengths:**

- The direction of self-rewarding is interesting, which makes the model being able to judge its own outputs and get progress.
- The idea of reducing dependency on high-quality data is attractive.

**Weaknesses:**

- Most of the ideas in this work seem to be from previous works and this work seems to focus on a combination of various methods. This is fine if this work aims to be an empirical one, however, the experiments and analyses are lacking, and there seems to be little insights on which factors work and why.
- There should be more baseline methods to prepare the proposed method, which is important to demonstrate certain choices of the proposed method.

**Questions:**

- I'm wonderin why specifically using ORPO? It seems that there are no direct comparisons to other methods such as DPO, more explanations and experimental results should be provided.

---

> ### Author Response · Authors · 2024-11-27
> **Response to Questions and Weaknesses**
>
> Thank you for your valuable feedback. We are grateful of your time invested in improving the quality of this research.
>
> The focus of this work is indeed empirical. We investigate whether multi task learning of alignment data, simultaneously with self-judgement data shall be able to provide improvements witnessed by approaches like iterative finetuning.
>
> To address your weaknesses:
>
> "There should be more baseline methods to prepare the proposed method, which is important to demonstrate certain choices of the proposed method."
>
> We have added another baseline approach of scalarization with Supervised Finetuning (SFT) for Self-judgement data (SJ) instead of ORPO loss that we used throughout our Multi-objective experiments. Like, epoch by epoch, and task by task approaches, it performs consistently below other works with a 13.1% win rate on Alpaca eval.
>
>
> To answer your question:  "I'm wondering why specifically using ORPO? It seems that there are no direct comparisons to other methods such as DPO."
> We choose ORPO because it improves upon DPO in performance as well as efficiency. DPO requires that Supervised Finetuning (SFT) and preference optimization be done in sequence, but ORPO method accomplishes that simultaneously. Our work similarly parallelizes self-judgement training with alignment training.

---

### Meta-Review · Area_Chair_yQ56 · 2024-12-12

**Metareview:**

The paper explores multi-objective optimization to align LLMs on answering and self-judging abilities by leveraging existing alignment data and ORPO as a baseline. It is innovative and promises efficiency improvements in LLM fine-tuning. However, all reviewers agree that this paper requires significant presentation, motivation, and experimental evaluation revisions. Therefore, I recommend rejecting this work.

**Additional Comments On Reviewer Discussion:**

The authors' response was delayed and insufficiently addressed key concerns.

While the authors describe their work as empirical, it lacks adequate experimental validation (reviewers 2Rx7, tkTp, LM5Q, and sjFv).

The evaluation is also insufficient (reviewers tkTp and LM5Q).

Consequently, the paper requires significant improvement before it can be considered for acceptance (reviewer aZDC).

Furthermore, reviewers 2Rx7 and LM5Q recommended using DPO as a baseline; however, the authors did not provide results for this comparison. At a minimum, they should have offered compelling evidence to satisfy the reviewers’ concerns.

Given these shortcomings, I recommend rejection.

---

### Decision · Program_Chairs · 2025-01-22

Reject